# Harnessing the Potential of Killers and Altruists within the Microbial Community: A Possible Alternative to Antibiotic Therapy?

**DOI:** 10.3390/antibiotics8040230

**Published:** 2019-11-21

**Authors:** Larisa N. Ikryannikova, Leonid K. Kurbatov, Surinder M. Soond, Andrey A. Zamyatnin

**Affiliations:** 1Institute of Molecular Medicine, Sechenov First Moscow State Medical University, Moscow 119991, Russia; surinder.soond@yandex.ru; 2Institute of Biomedical Chemistry, Pogodinskaya 10, Moscow 119991, Russia; kurbatovl@mail.ru; 3Belozersky Institute of Physico-Chemical Biology, Lomonosov Moscow State University, Moscow 119992, Russia

**Keywords:** bacterial allolysis, fratricide, cannibalism, microbial community, programmed cell death (PCD), toxins, bacteriocins, *Streptococcus pneumoniae*, *Bacillus subtilis*

## Abstract

In the context of a post-antibiotic era, the phenomenon of microbial allolysis, which is defined as the partial killing of bacterial population induced by other cells of the same species, may take on greater significance. This phenomenon was revealed in some bacterial species such as *Streptococcus pneumoniae* and *Bacillus subtilis*, and has been suspected to occur in some other species or genera, such as enterococci. The mechanisms of this phenomenon, as well as its role in the life of microbial populations still form part of ongoing research. Herein, we describe recent developments in allolysis in the context of its practical benefits as a form of cell death that may give rise to developing new strategies for manipulating the life and death of bacterial communities. We highlight how such findings may be viewed with importance and potential within the fields of medicine, biotechnology, and pharmacology.

## 1. Introduction

Bacterial allolysis is an interesting phenomenon which has been under discussion for the last 15 years [1,2,3,4,5,6,7,8,9]. It was firstly defined by Guiral et al. [10] to describe the phenomenon of cell demise in a subset of a bacterial population induced by other cells of the same species. The introduction of this special term was prompted by a necessity to differentiate this phenomenon from other types of cell lysis, such as heterolysis (or predation), which is directed against other unrelated species, or autolysis (which describes the self-destruction of cells). In the case of bacteria, the terms “fratricide” and “cannibalism” are often respectively associated with *Streptococcus pneumoniae* and other members of the *Streptococcus* genus and when speaking of “kin killing” in *Bacillus subtilis* populations [4,11,12,13]. The term “siblicide” is also used occasionally [14,15], as well as “sobrinicide”, to specify the type of killing observed between closely related species from the same phylotype [12].

Allolysis has been observed for a few species only (as mentioned above) and it has been suspected to occur in a case of some other respective genera and species like enterococci [16,17] or *Paenibacillus dendritiformis* [18,19]. It could be assumed that this phenomenon could arise and be related to the fundamental evolutionary processes inherent to all or at least most microorganisms. However, it is quite surprising that over the past two decades since the first mention of the kin killing phenomena that our knowledge still remains limited and within the paradigms presented by a few well-studied species. As was noted by Popp and Mascher in their very recent review [9], a fundamental shift and breakthrough in this area will probably require a change in the perception of microorganisms from single, planktonic cells to the conception of bacterial communities consisting of different sub-populations with distinct functions.

The biological role of allolysis is not yet completely understood and has raised a number of interesting questions for researchers. Certainly, the first question of interest is related to the purpose of this phenomenon—Why do cells kill their own siblings? The next question revolves around the mechanistic regulation of this process—How do “predator” cells that are successful in killing their isogenic neighbors confer resistance to this type of death, and what are the protective barriers? What is the basis by which the toxins are made by the “predators”? Are these toxins routinely used in normal processes within the cell or are they more specialized in their design, role, and specificity for killing siblings?

Indeed, the answers to such questions would certainly offer greater insight into how such processes are controlled and thus offer attractive prospects for managing the life and death of entire microbial communities—which is especially important now, in the era of “post-antibiotics” and antibiotic resistance. Consequently, in this mini-review, we summarize up-to-date findings and evidence to support the phenomena of allolysis in different bacterial species with a view to highlighting its emerging importance in medicine, with a particular emphasis on possibly harnessing this potential as an alternative to antibiotic therapy.

## 2. Kin Killing in *Lactobacillales*: Fratricide in *S. pneumoniae* and Enterococcal “Siblicide”

### 2.1. Fratricide in *Streptococcus*

In bacterial allolysis, the most characterized and studied model of cell death is the streptococcal fratricide phenomenon. As there are earlier excellent reviews on this topic which provide the basic principles of fratricide [4,6,12], we will try to bring into focus new findings obtained in recent years and extend the context of such findings. Generally, fratricide in *S. pneumoniae* is associated with a cascade of events linked to entering a state of competence closely linked to the genetic transformation process. Competence in *S. pneumoniae* can be triggered by some environmental and physiological signals, such as cell density, changes in pH, or the presence of antibiotics [8,20,21,22] during exponential growth, in a process that is governed by quorum sensing (QS) [22,23,24,25]. Changes in such conditions lead to an extracellular accumulation of the QS signal - the competence-stimulating peptide (CSP) or a pheromone, ComC. ComC is secreted from the cytoplasm by the ComAB transport system. Upon reaching an extracellular threshold concentration, it can interact with its cognate receptor at the cell surface, the transmembrane domain of ComD histidine kinase, leading to its activation. Consequently, activated ComD phosphorylates the cytoplasmic response regulator ComE, which in turn upregulates the expression of the 20 or so “early” competence (*com*) genes. One of these early *com* genes encodes the alternative sigma factor X (ComX) that activates the transcription of the “late” *com* genes, including the genes necessary for DNA binding, uptake, processing, and integration [26]. It was noted that the number of CSP-responsive genes in *S. pneumoniae* (over 100) are essentially excessive in the numbers required to permit the process of DNA transformation, suggestive of some being involved in other cellular processes beyond the molecular machinery required for DNA uptake and recombination [11,27].

Of relevance is one processes triggered by the induction of the competent state, which includes the expression of protein toxins (allolysis toxins or fratricines) that are deadly to neighboring sibling cells (Figure 1). To avoid the total death of the whole population, the protein that confers resistance (ComM) and provides protection against self-killing is also stimulated during the early stage of competence. Only the cells failing to become competent and therefore not protected by ComM undergo lysis by fratricide toxins [4,11,12]. The mechanistic reasons for why some of the cells do not respond to CSP signals, while the rest of the population develops the competent state, are not exactly known and form the basis of on-going research. However, it is well known that isogenic bacteria growing under identical conditions do not necessarily display the same pattern of gene expression during competence. This phenomenon is termed “bistability”, and is not associated with genome rearrangements or mutations, but alternatively involves reversible switching between varying states of competency in the absence of any genetic modification [12,23,28]. Consequently, it is possible that bistability gives rise to a mixed population of competent “predators” and non-competent target cells. Another potential source of heterogeneity is due to different streptococci cells producing different pheromones. Strains producing the same pheromones constitute the pherotype group. Pheromones can effectively induce a state of competence only within their pherotypes, so that within a mixed natural population of cells the members of one pherotype developing the state of competence the quickest can have a survival advantage over streptococci originating from other pherogroups, who are unable to accept CSP signals and thus remain non-competent [23]. In addition to bistability and pherotype diversity, the heterogeneous physico-chemical conditions existing inside pneumococcal colonies or within biofilms are believed to be important [16,17,29,30,31].

Experimentally, fratricide can be detected in liquid cultures based on the release of chromosomal DNA and cytoplasmic β-galactosidase as good indicators, as well as the prevalence of cell clumping [11,32,33,34,35], which also depends on the presence of extracellular chromosomal DNA [11]. Moreover, fratricide can be observed on solid blood agar plates, where the lysis of non-competent cells is observed with the peripheral appearance of a visible zone of β-hemolysis and which is due to the release of the cytolytic virulence factor pneumolysin (Ply) [10]. 

Concomitantly, some proteins have been identified as allolysis toxins: the ComX-dependent murein hydrolase choline-binding protein D (CbpD) and the autolysin LytA, as well as the competence-induced two-peptide bacteriocin CibAB (Table 1). Additionally, the non-CSP-regulated lysozyme (LytC) has been suggested to positively contribute to fratricide activity [30]. Structurally, CbpD (which is a 50-kDa product of the *cbpD* gene) contains a muralytic N-terminal CHAP (cysteine, histidine-dependent amidohydrolase/peptidase) domain followed by two SH3b (Src homology 3b) domains and a choline-binding domain made up of four choline-binding repeats at the C-terminal [30,36,37]. The choline-binding domain targets CbpD to the choline-decorated teichoic acids in the cell wall of the target cells, while the SH3b domains are thought to be involved in peptidoglycan binding, positioning the muralytic CHAP domain so that it can damage the cell wall of CbpD-susceptible cells. The presence of the choline-binding domain in this protein can be assumed to limit its cross-species activity to those cells who have choline in their cell wall, for example, closely related species of *Streptococcus* genus. In support of this, it was demonstrated that the pneumococcal CbpD could also trigger the release of DNA from closely related *Streptococcus mitis* and *Streptococcus oralis* [36,38].

LytA protein (36.5 kDa), the main pneumococcal autolytic enzyme and one of the most important pneumococcal virulence factors, also known as N-acetylmuramoyl-L-alanine amidase, is the product of the *lytA* gene. This protein degrades the peptidoglycan bonds of pneumococcal cell walls after attaching to the choline residues of the cell wall teichoic acids *via* its choline-binding module. It contributes to virulence probably by functioning as a releasing mechanism for the cytolytic toxin pneumolysin [39,40]. The 55 kDa autolysin LytC is a muramidase (lysozyme). It has no other known biological function except for its auxiliary role in CbpD-mediated lysis of susceptible cells [30,41,42,43].

CbpD plays a key role in mechanisms of cell lysis, as demonstrated by competence-induced cell lysis being abolished in the absence of this protein. It is suggested that CbpD introduces specific cuts into the peptide stems of pneumococcal peptidoglycan that allow other hydrolases, LytC and LytA, to become active resulting in more extensive lysis of target cells than that achieved by CbpD alone [10,42,44,45]. Whereas CbpD is expressed exclusively during competence, transcription of the *lytA* gene can be driven under several responsive conditions, one of which is competence inducible. Additionally, it is worth noting that the level of *lytC* transcription remains unaltered during the competent state [30,44]. 

As seen from early studies, it was established that the state of competence is associated with the production of bacteriocins, which are small bactericidal peptides characterized by their narrow spectra of inhibitory activity against diverse groups of microorganisms. The two-peptide bacteriocin CibAB was shown to participate in this process [10]. However, in the absence of other lytic enzymes (such as CbpD, LytA, and LytC), CibAB could not promote allolysis alone. The detailed mechanism of action of the CibAB toxin is largely unknown, but it has been suggested that it can act by inserting into the membrane of sensitive (non-competent) cells and depleting cellular energy, and thereby increasing their susceptibility to lysis. Competent cells can be protected by the immunity protein CibC [4,10,11]. Other bacteriocins, through varied regulatory pathways, may also be involved [46,47,48]. In their recent studies, two research groups independently described a mechanism for direct crosstalk between the *com* and *blp* (bacteriocin-like peptide) regulatory pathways [47,48]. It is known that the *blp* locus regulates the production of pneumococcal bacteriocin (pneumocin), and it does this in a manner similar to *com* regulation. Like competence, the *blp* locus is stimulated by the accumulation of a peptide pheromone (BlpC) that is processed and secreted out of the cell *via* its cognate transporter complex (BlpAB). BlpC activates the histidine kinase receptor`` BlpH, resulting in phosphorylation of the response regulator BlpR and upregulation of a variety of genes encoding bacteriocin and immunity [46,49,50]. Kjos et al. showed that the pheromone-induced two-component system driving competence for genetic transformation, ComDE, also controls expression of the “bacteriocin pheromone” BlpC. Moreover, in the conditions when the transporter complex of BlpC, BlpAB, was defective, the competence pheromone exporter ComAB could compensate to process and export BlpC in addition to the “competence pheromone” ComC [47]. Similar results were obtained by Wholey et al. [48], who demonstrated the induction of *blp* operon in response to exogenous CSP, and an important role of ComAB in the secretion of BlpC after CSP stimulation. Thus, the development of competence positively influences the regulation of the *blp* locus, allowing it to be speculated that the coordinated production of antimicrobial peptides during competence may provide pneumococcal strains an adaptive advantage by increasing a potential pool of DNA donors.

Different toxins seem to have a different biological impact, depending on the type of the assay applied [4,9]. Inactivation of CbpD had only a minor effect on pneumolysin release on plates [10], whereas it strongly affected DNA and cytoplasmic β-galactosidase release in liquid culture [45] and abolished clumping [11]. At the same time, bacteriocin CibAB did not contribute to the lysis of target cells in liquid cultures, but its inactivation led to the loss of pneumolysin release on plates [10,11]. Under biofilm conditions, efficient lysis of target cells required CbpD acting in combination with LytC, while autolysin LytA did not seem to be important for fratricide in the biofilm environment [29].

The immunity protein ComM is a 23.5-kDa integral membrane protein predicted to have six or seven transmembrane segments, short extra-membrane loops and protects streptococcal cells against CbpD through an unknown mechanism. ComM is a product of the early *com* gene, so its transcription level was reportedly elevated immediately after the induction of competence, peaking approximately 5 min after entering the competent state, which was just before the expression of the late *com* gene *cbpD* was initiated [37,51]. It was found that ComM induced during competence temporarily inhibited both the initiation of cell division and the final constriction of the cytokinetic ring. The ComMmediated delay in division may be necessary to preserve genomic integrity during transformation and/or to ensure that transformation is complete [51]. Overexpression of ComM results in growth inhibition and development of severe morphological abnormalities, such as cell elongation, misplacement of the septum, and inhibition of septal cross-wall synthesis. One of the hypotheses proposed for ComM-mediated immunity is that it acts by changing the cell wall structure of newly synthesized peptidoglycan within the septal area. It may modify a specific part of the peptide that is recognized and cleaved by the catalytic domain of CbpD. Also, ComM may introduce changes in peptidoglycan or teichoic acids which block the attachment of CbpD to the cell wall [37]. 

In addition to ComM, a second immunity factor CibC providing the protection against the two-peptide bacteriocin CibAB may also to be of importance. CibC is a putative transmembrane protein which is co-transcribed with CibAB, and its inactivation increases susceptibility of cells to allolysis [10].

Collectively, such findings highlight the importance of defining the transcriptional mechanisms underpinning the expression of the allolytic toxins, and future work may strive towards harnessing their potential in mediating cell death or their modes of action in future therapeutic design. 

### 2.2. Enterococcal “Siblicide”

A mechanism similar to streptococcal fratricide was also suggested for the phylogenetic neighbors of streptococci, namely enterococci. It is worth noting that in contrast to streptococcal fratricide, which has been extensively studied and characterized, the existence of such a phenomenon in the instance of enterococci is less clear and is rather a hypothesis deduced from a number of indirect observations. Here, an alternate hypothesis of altruistic suicide or any other form of programmed cell death may remain a possibility. The phenomenon of “siblicide” was observed in some clinical isolates of enterococci (*Enterococcus faecalis* and *Enterococcus faecium*), that, while growing as a colony (“stabs” into agar) on solid media, exhibited a growth-inhibitory effect on a lawn composed of the same strain [14]. This effect was attributed to the production of a class IIa bacteriocin (enterocin) (MC4-1) encoded on a conjugative, highly transferable, pheromone-responding, multiple antibiotic-resistance plasmid pAMS1 [14,52]. The lack of immunity against its own bacteriocin was explained by a difference in the expression levels of both the bacteriocin MC4-1 and corresponding immunity genes in the producer cells (a stabbed colony in early stationary phase growing under limited conditions) and target cells (a lawn in early exponential stage of growth). However, the bacteriocin-mediated nature of this phenomenon was questioned by the authors themselves, as some strains appeared not to be active against enterocin-sensitive *Listeria monocytogenes*. Consequently, the siblicidal phenomenon was not seen as exclusive to strains that produce class II bacteriocins [14,15,52].

In the *E. faecalis* model of fratricide and biofilm development proposed by Thomas and Hancock’s group [16,17,31,53], the molecular control of sibling killing depended on the regulation of QS, mediated by the two-component Fsr system. The *fsr* operon consists of four genes, *fsrABCD*, where *fsrA* and *fsrC* respectively encode the histidine kinase and its response regulator. The quorum signal encoded by *fsrD* and termed gelatinase biosynthesis-activating pheromone (GBAP) is an 11-amino-acid cyclized peptide lactone secreted extracellularly by means of FsrB. Approximately 10%–15% of cells in the stationary phase of an *E. faecalis* culture do not respond to the GBAP signal. Transcriptional profiling showed that GBAP-responder cells activated the expression of two co-transcribed and secreted extracellular proteases, the zinc metalloprotease (gelatinase) GelE, and the serine protease SprE. Relative to wildtype *E. faecalis*, isogenic ∆gelE mutants exhibited a reduced efficacy of autolysis while ΔsprE mutants contrastingly showed an increased rate of lysis, suggesting that GelE was probably playing a pro-lysis role while SprE acted in a lysis-inhibitory manner (Table 1). Mechanistically, it had also been suggested that GelE may act by modifying the surface of cells that do not respond to the GBAP signal, through modulating their interaction with AtlA, the main enterococcal autolysin [17,53] (Figure 1).

AtlA is one of at least three autolysins (AtlA, AtlB and AtlC) which have been identified to be secreted by *E. faecalis*. It is a soluble protein which is thought to be crucial for extracellular DNA (eDNA) release and biofilm formation. Based on sequence similarity, AtlA was proposed to be made up of three domains, with the central catalytic domain being responsible for the glucosaminidase activity. The C-terminal domain is composed of six LysM modules that afford peptidoglycan affinity and possibly target autolysins to the division septum and poles. No known function yet exists for the T/E-rich N-terminal domain [31]. It was deduced from cell wall zymography analysis that AtlA is a target of both of the proteases, GelE and SprE. The targeting of AtlA bound on the cell surface by GelE resulted in multiple AtlA enzymatically active forms, which caused the lysis of the GBAP non-responder cells. However, the role of SprE is less clear and this protein presumably assists in processing of AtlA to a discrete 62-kDa mature form that has a high affinity for cell walls and renders AtlA resistant to further proteolytic processing by GelE. Consequently, SprE may play the role of an immunity protein to protect GBAP responder cells from self-induced lysis. [16,17]. In general, the details of this interesting mechanism are still rather vague and need to be clarified further.

## 3. Sibling Killing in Bacillales: Cannibalism in *Bacillus subtilis* and Induced Suicide in *Paenibacillus dendritiformis*

### 3.1. Cannibalism in Bacillus subtilis

The allolysis-like phenomena in *B. subtilis* is associated with the sporulation process. When exposed to environmentally stressful conditions, for example starvation, this microorganism initiates a complex survival strategy which is called sporulation leading to the formation of endospores that are very resistant to adverse environmental conditions. It allows *B. subtilis* to survive and wait for favorable conditions for growth [9,28,54]. The entrance into the sporulation pathway is a multistep process associated with a diversification into distinct sub-populations of specialized cell types that try to extract different types of nutrients from the environment [55,56].

The sporulation process starts with the activation of the DNA-binding protein Spo0A, which is considered a key regulator of sporulation, and also biofilm formation and cannibalism [55,56,57]. This protein is activated by phosphorylation and controls the transcription of around 120 genes. A portion of these genes involved in biofilm formation and production of cannibalism toxins (see below) have high-affinity Spo0A binding sites and require only low levels of phosphorylated Spo0A (Spo0A-P), while other genes involved in the process of spore formation depend on higher Spo0A-P levels [58,59].

Sporulation is a very time and energy consuming process that becomes irreversible during certain stages of growth. In order to avoid this unprofitable burden, *B. subtilis* tries to delay sporulation as long as possible and employs different strategies to achieve this. One such “last delay” strategy (called cannibalism) involves the production and secretion of peptide toxins which are able to lyse sensitive siblings. Cannibalism toxins, as well as the immunity factors protecting the “cell-cannibals”, are probably produced by the sub-population of cells which have entered the sporulation pathway but have not yet completed it. The lysed cells are thought to provide nutrients for the cannibals to continue vegetative growth.

Among others, Spo0A regulates the transcription of two operons—*sdp* (sporulating delay protein) and *skf* (sporulating killing factor). These operons are highly expressed in cells expressing a low level of Spo0A [4], thus ensuring that their products are present at high levels in the early stages of the sporulation process. The eight-gene *skf* operon (*skfABCDEFGH*) is a typical antimicrobial peptide locus, where the first gene, *skfA*, encodes the 56-amino acid bacteriocin-like peptide which is post-translationally modified by a radical S-adenosyl-methionine enzyme SkfB and processed to its active state by the putative thioredoxin oxidoreductase SkfH [60,61]. SkfEF is the ABC transporter which exports SkfA out of cell and could be responsible for providing immunity against the mature SkfA toxin [13]. The *sdp* (*sdpABC*) operon encodes the toxic protein SdpC. Upon transcription, SdpC is post-translationally modified by SdpAB to its active 63-amino acid residue form. The mechanism by which this toxin acts is still unclear, but it may target the cell membrane causing it to become permeabilised and leaky [62]. It was demonstrated [63] that the SdpC cannibalistic toxin collapses the proton motive force and this subsequently induces autolysis in sensitive cells. SdpC also acts as a signaling molecule that induces the expression of the cognate autoimmunity gene spdI along with its repressor gene *sdpR* and therefore prevents suicide induced by SdpC. An autoregulatory feedback loop then ensures that Spo0A-active cells producing the toxin are protected from its damaging activity. Spo0A-inactive cells do not produce SdpC and hence are susceptible to the toxin [62,64].

With nutrient depletion, *B. subtilis* temporarily divides into sub-populations that differ in their levels of Spo0A-P expression. Those cells that reach the critical (but yet low) Spo0A-P level first, activate the “early” sporulation operons including the *skf* and *sdp* loci and produce the cannibalism toxins along with the immunity factors, lysing vegetative cells in which the Spo0A protein is still inactive. Herein, the products of the *skf* and *sdp* operons (at first, the immunity proteins) are still missing. As a result, the medium is temporarily enriched with substrates from killed vegetative cells providing a delay for spore formation in the toxin producing Spo0A-active sub-population. Furthermore, when these resources are also utilized, the concentration of Spo0A-P gradually increases and genes involved in the process of spore formation are switched on (Figure 2).

It is interesting that cannibalism toxins of *B. subtilis* are not highly specific to the producing strain. Supportingly, the Sdp toxin was demonstrated to be active against a variety of bacterial species inside the Firmicutes phylum, including both species closely (*Bacillus amyloliquifaciens*) or more distantly (*Lactobacillus acidophilus*) related to *B. subtilis*, or even unrelated species such as *Staphylococcus epidermidis* [63]. Experiments on the interspecies interactions between biofilms *B. subtilis* and related *Bacillus* species revealed that the cannibalism toxins secreted by *B. subtilis* upon interaction together with the peptide surfactin inhibited the growth and biofilm formation of *Bacillus simplex* and *Bacillus toyonensis* at concentrations that were inert to *B. subtilis* itself [65]. These findings demonstrated the important role of cannibalism in biofilm development in which cannibalism toxins can play a serious role in both intra- and interspecies competition, thus facilitating the survival of the *B. subtilis* population in biofilm communities [65,66]. 

### 3.2. “Forced Suicide” in Paenibacillus dendritiformis

*Paenibacillus dendritiformis* is an interesting bacterial species marked by the complex spatial organization of its colony, which can form different patterns of growth on semi-solid agar such as chiral branches, swirls, and vortices [67,68]. It was observed that the sibling *P. dendritiformis* bacterial colonies grown on low-nutrient agar medium mutually inhibited the growth of each other [69]. The factor responsible for the killing activity was purified from agar between growing competing colonies and named the “sibling-lethal factor” (Slf) (Table 1). Slf, a 12-kDa polypeptide, was shown to be derived from a precursor protein termed DfsB (dendritiformis sibling bacteriocin) which was cleaved by the serine protease subtilisin giving rise to its active form [18,19]. Subtilisin is another important factor unveiled to seemingly trigger the threshold response. It is secreted by *P. dendritiformis* into the environment and can be detected around the growing colonies, where its proteolytic activity may likely break down proteins to provide more accessible nutrient sources to the bacteria. As the concentration of subtilisin was proportional to the cell density it may therefore serve as a quorum sensor to control the growth of cells. When grown on low-nutrient agar medium (below a threshold concentration), subtilisin promoted colony growth and expansion. However, at the interface between competing colonies, the local concentration of subtilisin increased sharply, resulting in the secretion of an active form of Slf which lyses cells within the zone between the two colonies.

It was found that Slf was lethal to *P. dendritiformis* but not to other bacteria, even closely related ones, suggestive of the strict and narrow specificity of this toxin towards its own species [18]. It is also worth noting that while the production of Slf is induced by a sibling colony (via secreted subtilisin), the protein factor itself can be derived from the host, suggesting that this can be considered closer to induced suicide rather than direct fratricide [19].

Clearly, there is a need for further studies concerning this interesting bacterium, as it remains unclear what other growth conditions (besides starvation) cause the competitive capacity of single colonies to be manifested, or a detailed mechanism through which sensitive cells undergo cell death. Moreover, the most interesting question of how the bacterial cell protects itself from the death-inducing activity of its own toxin is also eagerly awaited.

## 4. Why do Bacteria Kill their Siblings?

This question is still under investigation and forms the basis of many on-going studies and the answer for which may lie in defining the diverse conditions under which bacteria are expected to thrive or survive. As for streptococcal fratricide, the answer mostly accepted by the scientific community is the one which offers a mechanism for cells to acquire homologous DNA from isogenic related bacteria during stress rather than a way to eliminate competing bacteria [4,11,12]. By having a large available gene pool, bacteria enjoy an evolutionary advantage by being able to adapt quickly to a changing environment. Another probable reason for fratricide is a necessity for the prompt release of cytoplasmic virulence factors (like pneumococcal pneumolysin) and inflammatory mediators from lysed cells during their interaction with the host, thus favoring pathogenicity [26]. At the same time, the factor of competition should not be completely excluded and has been addressed by Shen et al. [70], who studied the details of simultaneous colonization with multiple pneumococcal strains in vivo and showed that there is a competitive advantage for established resident strains over newcomers, even if the newcomer is isogenic to the resident strain. It was also established that *com*-regulated CbpD and CibAB bacteriocin (that mediate fratricide) play a central role. Here, the authors suggested that there is only a narrow window for co-colonization with different pneumococci strains which occurs before *com* activation by the resident strain, thus allowing general stability of the microbiome composition to be maintained over time.

In *Bacillus subtilis*, cannibalism is a strategy that allows cells to overcome temporary nutrient limitations and to postpone or potentially even prevent the onset of the unprofitable sporulation process under conditions of starvation. Resources released from killed relatives can be used by the toxin producers to continue with vegetative growth [5,9,13]. The protein toxin lethal to neighbor siblings is produced by *Paenibacillus dendritiformis* to regulate the density of their population and avoid overcrowding [18]. Moreover, the important role of the allolysis-like phenomenon in biofilm development has been shown for all species discussed herein [29,31,66]: indeed, the controlled lysis of bacterial sub-fractions results in the release of extracellular DNA, a crucial component of the biofilm matrix which contributes to the structural stability of the biofilm. Collectively, there is no doubt that such mechanisms are intimately associated with the adaptation of microbial communities to a change in environmental and growth conditions.

## 5. Allolysis as Weekdays of the Bacterial “Multicellular” Life 

Bacterial allolysis belongs to a class of programmed cell death (PCD) phenomena in prokaryotes. PCD is defined as an active, precisely regulated process for any form of cell death mediated by intracellular genetic programs [71]. Towards the end of the 20th century, it was discovered that some forms of PCD (including apoptosis) exists not only in multicellular animals, but also in plants, lower eukaryotes, and yeast cells [71,72,73]. PCD in prokaryotes remained questionable for a long time; however, the facts supporting this have been accumulating over time and which demonstrate that microorganisms often behave as members of a multicellular community, adding support to the idea that the analogues of PCD exist in bacteria [72,74,75,76].

Indeed, the ideas of altruistic self-killing or fratricide seem meaningless when applied to a single cell, as one cell cannot benefit from its own death. However, in their natural habitat, microorganisms do not usually exist as planktonic cells, they are social organisms and many of them can switch from unicellular to multicellular organization such as microbial colonies, biofilms and aggregates. “Multicellular” life styles can offer many advantages, such as increased protection against hostile environments, increased genetic diversity and improved food availability and is characterized by clear lines of communication between the members (e.g., quorum sensing) and differentiation into the specialized subpopulations (“the division of labor”) as well as complex patterns of collective behavior (shared hunting, collective resistance to antibiotics, etc.) [72,77]. In the “multicellular” bacterial community, sacrificing part of the population can be a way to sustain the survival of the remaining cells.

Irrespective of the perspective, studies on this aspect of “multicellular” bacterial life are of great interest because they may be exploited for developing novel therapeutic strategies against pathogenic bacteria in the era of “post-antibiotics”, antibiotic resistance, and the emergence of “superbugs”, which all pose a threat to human life. Therefore, as an alternative to killing single bacterial cells through the use of antibiotics, it may be realistic to concentrate efforts on bacterial killing through the disruption of the bacterial community as a whole. Such an approach has been developed as a potential alternative to antibiotics for a significant period of time. Antagonists to the bacterial quorum sensing system (anti-QS agents), which are aimed at specifically disrupting intercellular signaling within the bacterial community and which permit sensitivity to the human immune system, are ongoing areas of research with a realistic impact [78]. While the clinical applications of anti-QS agents are still developmental in their nature, the potential to disrupt communication between cells as well as activating self-killing within bacterial communities could be used for therapy in the future. With this in mind, the idea of utilizing and targeting bacterial suicide pathways within bacterial communities does offer realistic and attractive clinical alternatives.

## 6. The Inter-Relationships within the Microbial Community—Unfolding the Puzzle

The idea of “multicellularity” within microbial communities opens novel aspects for understanding the prokaryotic world. Each novel piece of work within this field adds a new element to the puzzle, thus gradually revealing the whole picture—a picture of complexity and multi-levelled interactions within the bacterial community. Of focus within this picture are: (1) the systems of cell-to-cell communication, including those in growth conditions of stress, as well as mechanisms of kin recognition [79]; (2) a distribution of duties inside the “microbial cities”; and (3) a rational utilization of resources and the ergonomic regulation of the inter- and intra-cellular processes [28,57]. As highlighted above, allolysis is a variant of the bacterial PCD and forms only part of the comprehensive picture that is unfolding. Consequently, of interest would be whether this phenomenon is inherent in all members of the microbe world or whether it occurs exclusively in just a few species. Is there any common basis for the purpose and need for this process in between different microorganisms, or is each case unique? Actually, while the fratricidal CibAB toxin loci are widely distributed, the homologs of the cannibalism toxins’ (Sdp and Skf) genetic loci are rarely found outside of the *B. subtilis* genome, indicating a very unique role for these proteins in the life cycle of this species [9]. However, it has been shown that the *dfsB* gene encoding the precursor protein DfsB, which is further processed to the *P. dendritiformis* sibling lethal factor (Slf), is widely distributed across gram-positive bacteria and to some extent gram-negative bacterial kingdoms [18,19]. This suggests that this mechanism for self-regulation of population growth might not be limited to *P. dendritiformis* only. In streptococci, many of the features discovered in *S. pneumoniae* are thought to be resident in other members of the genus *Streptococcus*, at least within the *mitis* group. In support, major oral streptococcal species such as *Streptococcus gordonii*, *Streptococcus sanguinis*, and *Streptococcus mutans* are naturally competent and have a range of autolysin homologues that can significantly contribute to the lysis of siblings [49,80,81,82,83,84,85]. 

In summary, among all the fragmented findings highlighted herein, the major inference is that the resulting feature of this kind of bacterial anti-proliferative phenomenon permits phenotypic heterogeneity and the diversification of the isogenic bacterial population—either permanently or transiently [9].

## 7. Conclusions

Bacterial allolysis is an important phenomenon for which many scientific aspects are yet to be clarified. A considerable difficulty in studying the details of this phenomenon probably arises from the inability to categorize it further into sub-forms of PCD, such as autolysis or altruistic suicide, or conversely as predatory. However, bringing into focus the broader picture that clearly defines the inter-relationships between the members of this complex “multicellular” organism (known as the bacterial community) can serve as a strong basis for unveiling fundamental new insights into prokaryotic life. Such insights may not only have a theoretical impact, but may also offer great practical and applied benefits in medicine, biotechnology, and pharmacology.

## Figures and Tables

**Figure 1 antibiotics-08-00230-f001:**
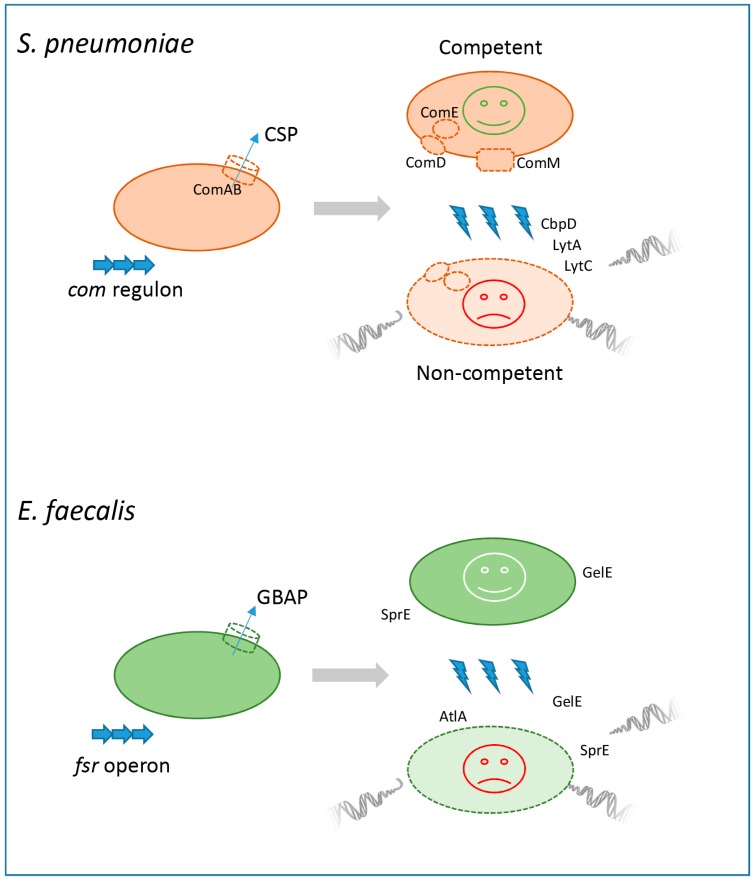
Fratricide in *Streptococcus pneumoniae* and enterococcal “siblicide”.

**Figure 2 antibiotics-08-00230-f002:**
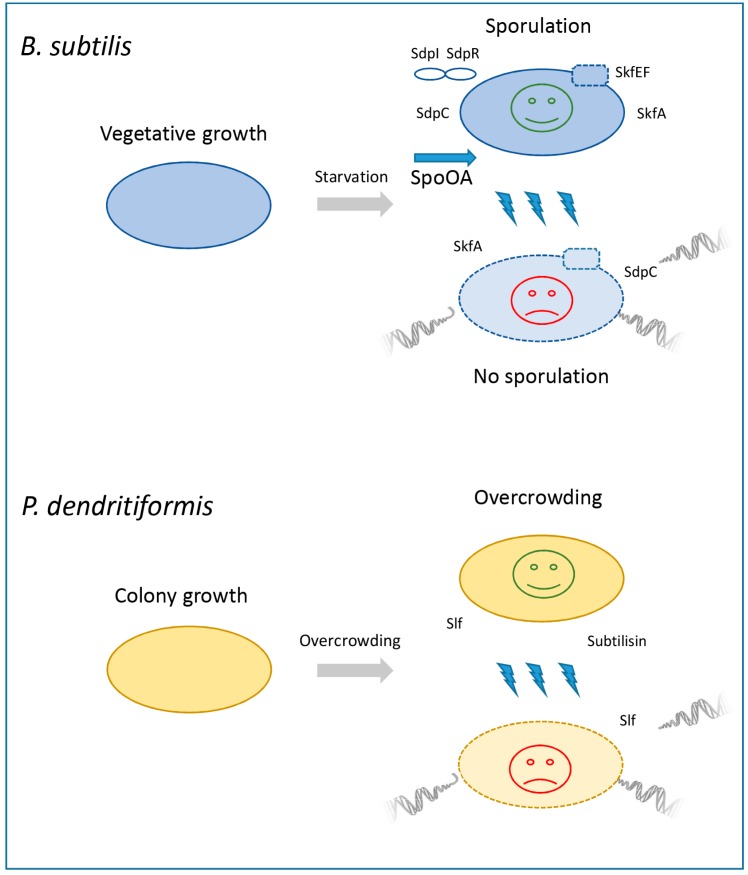
Sibling killing in Bacillales.

**Table 1 antibiotics-08-00230-t001:** Toxins and immunity factors in allolysis. CSP: competence-stimulating peptide.

Microorganism	Regulation	Toxin		Role
*S. pneumoniae*	ComX-dependent	CbpD	Murein hydrolase choline-binding protein D	Fratricine
	ComX-dependent	LytA	N-acetylmuramoyl-L-alanine amidase	Autolysin
	non-CSP-regulated	LytC	Muramidase (lysozyme)	Autolysin
		ComM	Membrane protein	Immunity
	ComX-dependent	CibAB	Two-peptide bacteriocin	Auxiliary role in fratricide
		CibC	Putative transmembrane protein	Immunity
*E. faecalis*	GBAP-dependent	GelE	Zinc metalloprotease (gelatinase)	Pro-lysis role
		AtlA	N-actyl glucosaminidase	Autolysin
	GBAP-dependent	SprE	Serine protease	Putative immunity function
		MC4-1 (product of *bacA* gene)	Plasmid-encoded class IIa bacteriocin	Bacteriocin
		BacB	Putative bacteriocin immunity protein	Immunity
*Bacillus subtilis*	SpoOA-regulated	SkfA	Bacteriocin-like pro-peptide	Bacteriocin
		SkfEF	ABC transporter	Putative immunity function
	SpoOA-regulated	SdpC	Toxic protein	Toxin
		SdpI		Immunity
*Paenibacillus dendritiformis*		Slf	Peptide sibling-lethal factor	Toxin
		Subtilysin	Serine protease	Auxiliary role

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
