# Peer review of "Harnessing the Potential of Killers and Altruists within the Microbial Community: A Possible Alternative to Antibiotic Therapy?"

_antibiotics, 2019, doi:10.3390/antibiotics8040230_

Round 1

Reviewer 1 Report

In this review article, entitled “Harnessing the potential of killers and altruists within the microbial community: a potential alternative to antibiotic therapy?” the authors are trying to understand the phenomenon of microbial allolysis i.e., self lysis of a part of the bacterial population and its importance in maintaining the bacterial communities.   

In general, the manuscript is well-written; however, this manuscript has a few weaknesses that could have been addressed.

For example, it is not clear how the bacteriocidal peptides insert itself into the membrane of non-competent cells?

What are the molecular mechanisms by which competent cells recognize non-competent cells?

How this phenomenon could be applied to benefit the field of medical science?.

Author Response

We are very grateful to the reviewers who generously donated their time to review our manuscript.

We have addressed all of the points raised for the peer-review process (in a point-to-point manner) and our responses are pasted below. We have made the requested alterations to the text of the manuscript (using tracked changes) and we do believe that it helps the reader as a result of these changes through enhanced clarification and fluency.

It is not clear how the bactericidal peptides insert itself into the membrane of non-competent cells?

As outlined in the text “However, in the absence of other lytic enzymes (such as CbpD, LytA and LytC), CibAB cannot promote allolysis alone. This suggests that this peptide supports cell lysis by possibly inserting itself into the membrane of non-competent cells and de-energizing them, thereby increasing their susceptibility to lysis”, lines 151-154 of uncorrected version), because the mechanisms of action of other toxins involved in streptococcal fratricide are described quite detailed in p. 4 (lines 120-150). However, in the case of the CibAB bacteriocin, there is very little published regarding its mode of action, outside of what is detailed by Guiral (ref. 10) and who suggest (referring to the van Belkum et al., J. Bacteriol. 1991, 173, 7934) that it can act by inserting into the membrane of sensitive cells and depleting cellular energy.

Consequently, we have altered how this section reads by changing it to “The detailed mechanism of action of the CibAB toxin is largely unknown, but it has been suggested that it can act by inserting into the membrane of sensitive (non-competent) cells and depleting cellular energy, and thereby increasing their susceptibility to lysis. Competent cells can be protected by the immunity protein CibC.” (lines 154-159).

What are the molecular mechanisms by which competent cells recognize non-competent cells?

From intensely perusing the literature, the specific mechanism for recognition of non-competent cells by competent ones remains to be elucidated and is a relatively unexplored area of work. For clarification, non-competent cells are induced to undergo lysis by competent cells because they are not protected by the immunity protein whose production is initiated prior to the appearance of the fratricidal toxins. So, the transition to a competent state involves the sequential production of immunity protein and (some time later) the production of toxins. This point has been detailed in the text from lines 90-93 (“To avoid the total death of whole population, the protein that confers resistance (ComM) and provides protection against self-killing is also stimulated during the early stage of competence. Only the cells failing to become competent and therefore not protected by ComM undergo lysis by fratricide toxins”) and lines 194-197 (“ComM is a product of the early com gene, so its transcription level was reportedly elevated immediately after the induction of competence, peaking approximately 5 min after entering the competent state, which was just before the expression of the late com gene cbpD was initiated”).

The same is true in respect of the CibAB bacteriocin. As suggested by its name (“Competence-Induced Bacteriocin”), regulation of the cibABC genetic locus is competence-dependent and competent pneumococcal cells expressing the CibAB toxin along with the immunity protein (CibC) target nearby non-competent pneumococci cells that are unable to express immunity protein. The text has been modified to reflect this point more clearly (see previous point and lines 154-159).

How this phenomenon could be applied to benefit the field of medical science?

Thank you for bringing this point to our attention. As the focus of this review looks at the potential application of allolysis in medicine, we think this point is addressed in a balanced manner and is supported by the limited literature that has been published in this context to date. We belive that introducing a new section to address the medical applications excessively would alter the tone of the manuscript excessively to be ‘too hypothetical’ and ‘speculative’. Consequently, we have expanded the discussion at the end of chapter entitled “Allolysis as weekdays of the bacterial “multicellular” life” (lines 414-423) to address this point by placing greater emphasis on modulation of the “life and death” of bacterial communities as a way of controlling human infectious diseases (particularly ones that are untreatable with antibiotics). Additionally, methods directed at disruption of communication between single cells within the bacterial community (like blockers of quorum sensing signals) and which make it susceptible to the host immune system are currently part of ongoing studies, with the practical applications of allolysis being a topical and relatively new area of study. For example, a project outline aimed at characterising substances capable of inhibiting ComX-regulated allolysis in streptococci can be found at the following url link: http://grantome.com/grant/NIH/R01-HL142626-01A1.

Reviewer 2 Report

The mini-review "Harnessing the potential of killers and altruists within the microbial community: a potential alternative to antibiotic therapy?" effectively summarizes an interesting phenomenon of microbiology: "siblicides" in microbial community. From the microbiological point of view, the review focuses on the salient points of this phenomenon with an adequately updated bibliography. The manuscript is readable and the English style is fine. Considering the molecular aspects described and the importance of this phenomenon not yet well characterized in the field of microbiology, the manuscript can be considered suitable for publication after very minor revisions: 

-I would avoid repeating the word "Potential" in the title, e.g. replacing the second with "possible" or something like that.

-Please check that all the names of the bacterial strains are in italics.

Author Response

We are very grateful to the reviewers who generously donated their time to review our manuscript.

We have addressed all of the points raised for the peer-review process (in a point-to-point manner) and our responses are pasted below. We have made the requested alterations to the text of the manuscript (using tracked changes) and we do believe that it helps the reader as a result of these changes through enhanced clarification and fluency.

I would avoid repeating the word "Potential" in the title, e.g. replacing the second with "possible" or something like that.

Thank you for this oversight. The wording of the title has been altered (as recommended).

Please check that all the names of the bacterial strains are in italics.

All names of strains as well as the names of genetic loci have been thoroughly checked and reformatted using italics (see lines 25, 65-66, 81, 124, 132 etc.).

Additionally, some other misprint were also corrected (it is highlighted in the text by the "Track Changes" function).

The minor revision of English was also made, to make it more clear.

One more reference (ref. 78) was added. The numbers of the other references have been changed accordingly.

Round 2

Reviewer 1 Report

The authors have addressed my concerns and I do not have any further comments on this manuscript.